# Numerical rainfall simulation with different spatial and temporal evenness by using WRF multi-physics ensemble

Jiyang Tian[1], Jia Liu[1,2], Chuanzhe Li[1], Fuliang Yu[1]

[1]State Key Laboratory of Simulation and Regulation of Water Cycle in River Basin, China Institute of Water Resources and Hydropower Research, Beijing, 100038, China
[2]State Key Laboratory of Hydrology-Water Resource and Hydraulirc Engineering, Hohai University, Nanjing, 210098, China

*Correspondence to*: Jia Liu (hettyliu@126.com)

**Abstract.** The Weather Research and Forecasting (WRF) model is used in this study to simulate six storm events in two semi-humid catchments of Northern China. The six storm events are classified into four types based on the rainfall evenness in the spatial and temporal dimensions. Two microphysics, two planetary boundary layers (PBL) and three cumulus parameterizations are combined to develop an ensemble containing 16 members for rainfall generation. The WRF model performs the best for Type 1 event with relatively even distributions of rainfall in both space and time. The average relative error (ARE) for the cumulative rainfall amount is 15.82%. For the spatial rainfall simulation, the lowest root mean square error (RMSE) is found with event II (0.4007) which has the most even spatial distribution, and for the temporal simulation the lowest RMSE is found with event I (1.0218) which has the most even temporal distribution. It is found to be the most difficult to reproduce the very convective storm with uneven spatiotemporal distributions (Type 4 event) and the average relative error (ARE) for the cumulative rainfall amounts is up to 66.37%. The RMSE results of Event III with the most uneven spatial and temporal distribution are 0.9688 for the spatial simulation and 2.5327 for the temporal simulation, which are much higher than the other storms. The general performance of the current WRF physical parameterizations is discussed. The Betts-Miller-Janjic (BMJ) is found to be unsuitable for rainfall simulation in the study sites. For Type 1, 2, and 4 storms, member 4 performs the best. For Type 3 storms, member 5 and 7 are the better choice. More guidance is provided for choosing among the physical parameterizations for accurate rainfall simulations of different storm types in the study area.

## 1 Introduction

Precipitation is a crucial element in the hydrological cycle at regional or global scales. With the characteristics of high intensity, short duration, uneven distribution and sudden occurrence, the precipitation easily causes flood with high peak in semi-humid region, which is tricky for forecasting accurately (Nikolopoulo et al., 2010). Quantitative precipitation forecast (QPF) is an effective method to avoid flood disasters and help flood risk management (Kryza et al., 2013). With the development of computer technology and atmospheric physics, numerical weather prediction (NWP) has become an efficient method for QPF (Yang et al., 2012).

As the latest generation mesoscale NWP system, the Weather Research and Forecasting (WRF) model can apply to the regions across scales from tens of meters to thousands of kilometers. Not only the rainfall quantity but also the spatial and temporal patterns of rainfall can be captured by WRF model with high resolution. Though it has been confirmed by many studies that the WRF model performs better than the Fifth Generation Penn State/NCAR Mesoscale Model (MM5), rainfall

is still one of the most difficult variables to simulate and predict (Collischonn et al., 2005; Bruno et al., 2014; Junhong et al., 2015). Because of the complicated processes of storm formation and development, the WRF model provides various physical parameterizations to be applied in different cases. Each physical parameterization emphasizes on different physical processes and has its unique structure and complexity, which may have great influence on the rainfall simulations. That is why numerous sensitivity studies of the WRF parameterizations are carried out in different regions of the world (Klein et al.,

2015). Three categories of the parameterizations have been mostly discussed and identified as the main influencing factors for rainfall simulation, i.e., microphysics, planetary boundary layer (PBL) and cumulus parameterization. Different physical parameterizations are found to be efficient for different rainfall event in different region (Jankov et al., 2011; Madala et al., 2014; Pennelly et al., 2014).

It is an increasingly difficult task to determine the optimal combination of physical parameterizations as the development of

the WRF model with more and more choices of parameterizations. Although many studies show that the best physical parameterization combination can be found by a lot of simulations for a certain rainfall event, it is difficult to tell the characteristics of the future rainfall events for real-time rainfall prediction. In order to consider the uncertainties associated with the selection of physical parameterizations, it has become a common method to use the ensemble in numerical rainfall prediction (Evans et al., 2011). Flaounas et al. (2011) studied an ensemble with six members over the west Africa, which

was produced by two PBL and three cumulus parameterizations. An ensemble containing 18 members was investigated in south-central United States, which was created by three microphysics, three PBL and two cumulus parameterizations (Jankov et al., 2005). And an ensemble with 36 members was tested for a series of rainfall events at the south-east coast of Australia, which contained two PBL, two cumulus, three microphysics and three radiation parameterizations (Flaounas et al., 2011). These studies show that no single physical parameterization combination performs the best for all rainfall events.

In this study, 16 physical parameterization combinations are designed from two microphysics of Purdue-Lin (Lin) and WRF Single-Moment 6 (WSM6), two PBLs of Yonsei University (YSU) and Mellor-Yamada-Janjic (MYJ), and three cumulus parameterizations of Kain-Fritsch (KF), Grell-Devenyi (GD) and Betts-Miller-Janjic (BMJ). Lin is a sophisticated parameterization which contains five classes of hydrometeors, and it is suitable for high-resolution simulations (Lin et al., 1983). WSM6 reveals an improvement in the high cloud amount and surface precipitation, which adds graupel microphysics

based on the works of Lin et al. (1983) and Rutledge and Hobbs (1983). MYJ PBL is appropriate for all stable or slightly unstable flows (Janjic, 1994). YSU PBL improves the performance of intense convection based on the Medium Range Forecast (MRF) PBL (Hong et al., 2006). KF is a classic cumulus parameterization and has been used successfully for years in many scientific institutions (Kain, 2004). GD is an ensemble cumulus parameterization and can be used in high resolution

models (Grell and Freitas, 2014). BMJ can adjust instabilities in the environment by generating deep convection and has been used extensively throughout the globe (Janjic, 2000).

Two medium sized catchments Fuping and Zijingguan are chosen as the study sites, which respectively locates in the south and the north reaches of Daqinghe catchment in North China. With the characteristics of high intensity, short duration, uneven distribution and sudden occurrence, the storm events in the study sites are representative for the semi-humid region with temperate continental monsoon climates. The aim of this study is to find out the potential performance of the WRF model for different types of storm events in semi-humid regions. Six storm events are chosen from the study sites and classified into four different types based on the rainfall evenness in the spatial and the temporal dimensions. The 16 designed combinations of physical parameterizations are treated as the ensemble for rainfall simulation and the results are verified regarding both the cumulative rainfall amounts and the spatiotemporal patterns.

## 2 WRF model configuration and designed physical ensemble

The 3.6 version of the WRF model is used in this study. WRF is a fully compressible, nonhydrostatic, meteorological model and it features physics, numerics, advanced dynamics and data assimilation. The model manual (Skamarock and Klemp, 2008) shows more detailed information of the WRF model. Two-way nesting is allowed for the communication between multiple domains at different grid resolutions, and three nested domains are centred over the Fuping and Zijingguan catchments respectively. In general, high-resolution rainfall products downscaled by the WRF model are more appropriate to be used as the input of the hydrological models (Cardoso et al., 2013; Chambon et al., 2014). So horizontal grid spacing of the WRF innermost domain is set to be 1km and the downscaling radio is set to be 1:3 (Givati et al., 2012; Yang et al., 2012). The centre of the domain is at lat 39°04′15″N and long 113°59′26″E, and the nested domains sizes are 252×234, 144×126 and 96×84 km$^2$ for Fuping catchment. The centre of the domain is lat 39°25′59″N and long 114°46′01″E, and the nested domains sizes are 216×198, 108×90 and 72×42 km$^2$ for Zijingguan catchment. The nested domains and the orography of the two catchment are showed in Fig.1. There are 40 vertical levels for three domains and the top level is set at 50hPa (Aligo et al, 2009; Qie et al, 2014). The WRF model is initialized from the six-hourly global analysis data provided by the 1°×1° grids of the NCEP Final (FNL) operational model. The integration step of WRF follows the '6×dx' rule where dx is the grid spacing, and the integration step is 6s for innermost domain (Skamarock and Klemp, 2008). The time step of WRF model output is set to one hour. The spin-up period is necessary for WRF model to develop the smaller scale convective features and the widely used lengths are 6 hours (Givati et al, 2012), 12 hours (Hu et al, 2010) and 24 hours (Wang et al, 2012). Different spin-up lengths were tried for the six storm events in this study, whereas results did not show obvious differences regarding the simulated rainfall. In order to improve the calculation efficiency for further hydrological use (i.e., flood warning), a 6h period is chosen to spin-up the model. That is to say the start of the model integration is 6 h earlier than the storm start time and the end time of the model integration is consistent with the storm end time.

The setting of the WRF model is very important before it is used to simulate the meteorological factors, especially the physical parameterizations. As shown by Table 1, a WRF physical ensemble is constructed by combining different choices of the physical parameterizations to simulate the storm events in the study areas. The selection of the parameterizations is based on their good performance in semi-humid region of China (Givati et al., 2012; Qie et al., 2014; Di et al., 2015). In order to learn the physical parameterizations more comprehensively, the different complexity and mechanism are also considered. WSM6 is the most complex in the series of WSM schemes, which is revised based on Lin (Hong and Lim, 2006). YSU is a non-local closure scheme whilst while MYJ is a local-closure scheme (Evans et al., 2011). The KF has a simple cloud model which can be triggered when air parcels temperature at its lifting condensation level is larger than the environmental air (Pennelly et al., 2014). The GD can run effectively within each high resolution grid (Grell and Freitas, 2014). The BMJ scheme is more suitable for convective weather because it can adjust the model profile of temperature and moisture (Janjic, 2000). Some studies indicated that the cumulus parameterizations may be invalid with fine horizontal resolutions, while the threshold of the resolution is unknown (Argüeso et al, 2011; Evans et al, 2012; Pei et al, 2014). Many studies use cumulus parameterizations with the about 1km resolution for weather simulation. For example, Shepherd et al (2016) explored the sensitivity of hurricane track to four cumulus parameterizations, including KF, BMJ, G-3 and TD, with the nested domains 1.33km, 4km and 12km. Remesan et al (2015) studied the WRF model sensitivity to the choice of parameterizations: 4 nested domains (1km, 3km, 9km and 27km) are used and the cumulus parameterizations of GD, BMJ, KF1 and KF2 are investigated. In order to make the study more rigorous, member 13, 14, 15 and 16 are also tested and compared with the members containing cumulus parameterizations. Many studies indicate that the simulation of precipitation is insensitive to the land-surface model (LSM), short and long-wave radiation parameterizations, so Noah for LSM, RRTM/Dudhia for long/short radiation are used in this study, which are most frequently applied to precipitation simulation (Guo et al., 2013; Chen et al., 2014).

[Figure 1 and Table 1]

## 3 Storm events and evaluation statistics

### 3.1 Study area and storm events

Fuping and Zijingguan catchments are the study areas, which respectively belong to the south and north reaches of the Daqinghe catchment, located in Northern China with semi-humid climatic conditions. The drainage area of Fuping (from lat 39°22′ to lat 38°47′N and from long 113°40′ to long 114°18′E) is 2210km$^2$ and the area of Zijingguan (from lat 39°13′ to lat 39°40′N and from long 114°28′ to long 115°11′E) is 1760km$^2$ (shown by Fig. 2). The average annual rainfall is about 600 mm at the study sites and the majority of rain focuses in the flood season. As shown by Fig. 2, there are 8 rain gauges in Fuping catchment and 11 rain gauges in Zijingguan catchment. The observed hourly rainfall data from rain gauges are

treated as the ground truth. Six 24-h storm events are selected from the recent 10 years (2006 to 2015) with respective rainfall characteristics of the study sites. The encounter between the western pacific subtropical high and the cold vortex of westerlies, and strong upward motion caused by Taihang Mountain are the main factors of rain formation in the study area. While the six storm events have quite different spatial and temporal evenness. Table 2 shows the duration and accumulative rainfall amounts of the six storm events.

**[Figure 2 and Table 2]**

The six storm events are categorized into four types based on the rainfall evenness of the spatiotemporal distribution (Liu et al, 2012). The variation coefficient $C_v$ is used to evaluate the uneven level:

$$C_v = \sqrt{\frac{1}{N} \sum_{i=1}^{N} (\frac{x_i}{\overline{x}} - 1)^2} \qquad (1)$$

For the spatial distribution, $x_i$ is the 24-h rainfall accumulation at rain gauge $i$, and $\overline{x}$ is the average of $x_i$. N is the number of rain gauges. For the temporal distribution, $x_i$ is the hourly areal rainfall at time $i$, and $\overline{x}$ is the average of $x_i$. N is the number of hours.

The higher of $C_v$, the more uneven for rainfall. In order to learn the spatial and temporal evenness of the rainfall in the two catchments, both spatial and temporal $C_v$ of the storm events from 1985 to 2015 are all calculated. In reality, rainfall in Northern China is much more uneven than the south and it is impossible to find absolute even rainfall in both space and time. So we chose a threshold of 5%, which is also considered in other statistical analyses in the same area, as the critical value to separate even and uneven rainfall events. With the threshold, we found the two critical values of 0.4 for the spatial $C_v$ and 0.6 for the temporal $C_v$. That is to say, the storm events with the spatial $C_v$ below 0.4 or with the temporal $C_v$ below 1.0 account for 5% of the total storm events from 1985 to 2015 in the study area. Table 3 shows the spatial and temporal $C_v$ of observations for the six storm events. Storm type 1 is characterized by even spatiotemporal distribution of rainfall. For storm type 2, rainfall is even for spatial distribution, but the temporal distribution is uneven. Storm type 3 and type 4 are characterized by an uneven distribution of rainfall in both space and time, while the rainfall of type 4 is highly concentrated in space and time. Due to the temperate continental monsoon climate in the study sites, there is no storm event with rainfall even and continuous in time but unevenly distributed in space.

**[Table 3]**

## 3.2 Verification indices for rainfall simulations

For evaluating the accuracy of rainfall simulation, both the accumulated areal rainfall and the spatiotemporal distribution of the rainfall are important. The accumulated areal rainfall is evaluated by the relative error (RE):

$$RE = \frac{(P - Q)}{Q} \times 100\% \qquad (2)$$

where $P$ is the simulated value, which is the average value of all the grids inside the study area; $Q$ is the observed value, which is calculated by the Thiessen polygon method based on the observations of the rain gauges (Sivapalan and Blöschl, 1998; Jarvis et al, 2013).

The spatial and temporal distributions of the rainfall are evaluated by a two-dimensional verification scheme. Both in spatial and temporal dimensions, some categorical and continuous indices are selected and calculated (Liu et al., 2012). The categorical verification indices are chosen as the probability of detection (POD), the frequency bias index (FBI), the false alarm ratio (FAR) and the critical success index (CSI). The calculation of the categorical indices depends on whether it rains or not, as shown in Table 4. It should be mentioned that the insignificant precipitation (less than 0.1 mm/h) is regarded as no rain. For verification in the spatial dimension, the comparison is made between the observations of the rain gauges and the simulations of WRF model at each time step $i$, and then the average values are calculated by the categorical indices at all the time steps as the final results. As shown by the Eq. (3)-(6), N is the total number of time steps of the WRF model output, which is 24 in this study. For the temporal dimension, the time series data of simulation and observation are used to calculate the four indices at each rain gauge $i$, then the average values are calculated by the indices at all the rain gauges as the final results. This time N is the number of the rain gauges of Fuping and Zijingguan catchments respectively in Eq. (3)-(6).

**[Table 4]**

$$POD = \frac{1}{N} \sum_{i=1}^{N} \frac{NA_i}{NA_i + NC_i} \qquad (3)$$

$$FBI = \frac{1}{N} \sum_{i=1}^{N} \frac{NA_i + NB_i}{NA_i + NC_i} \qquad (4)$$

$$FAR = \frac{1}{N} \sum_{i=1}^{N} \frac{NB_i}{NA_i + NB_i} \qquad (5)$$

$$CSI = \frac{1}{N} \sum_{i=1}^{N} \frac{NA_i}{NA_i + NB_i + NC_i} \qquad (6)$$

For the four categorical indices, POD indicates the percentage of correct simulation for the observed rainfall. FBI shows whether the WRF model has a tendency to overestimate (FBI>1) or underestimate (FBI<1) rainfall occurrences, while FBI cannot show closeness of the simulation and the observation. FAR represents the ratio of false alarms, and CSI indicates the

percentage of correct simulation between the simulated and observed rainfall. The perfect scores of POD, FBI, FAR and CSI are 1, 1, 0 and 1, respectively.

Besides the categorical indices, three continuous indices including the root mean square error (RMSE), the mean bias error (MBE) and the standard deviation (SD) are adopted for more quantitative evaluation of the simulated rainfall distributions in space and time. The calculations of the three continuous indices are expressed by Eq. (7)-(9).

$$RMSE = \frac{\sqrt{\frac{1}{M}\sum_{j=1}^{M}(P_j - Q_j)^2}}{\frac{1}{M}\sum_{j=1}^{M}Q_j} \times 100\% \tag{7}$$

$$MBE = \frac{\frac{1}{M}\sum_{j=1}^{M}(P_j - O_j)}{\frac{1}{M}\sum_{j=1}^{M}Q_j} \times 100\% \tag{8}$$

$$SD = \frac{\sqrt{\frac{1}{M-1}\sum_{j=1}^{M}(P_j - O_j - MBE)^2}}{\frac{1}{M}\sum_{j=1}^{M}Q_j} \times 100\% \tag{9}$$

For the spatial dimension, $P_j$ and $Q_j$ are the simulation and observation of 24 h rainfall accumulations at each rain gauges $j$. M is the number of the rain gauges, which is 8 for Fuping catchment and 11 for Zijingguan catchment. For the temporal dimension, $P_j$ and $Q_j$ are the average areal rainfall simulation and observation at each time step $j$. This time M is 24 which represents the number of the time steps. The final values of the three indices represent the mean magnitude of error, the average cumulative error and the variation of the simulation error of MBE, respectively. The perfect score of all the three indices is 0. In order to compare the simulations for different storm events, the final values of the three continuous indices in both two dimensions are represented as percentages of the corresponding average observations.

## 4 Results

### 4.1 Simulations of the 24h rainfall accumulations

The simulation results of the cumulative rainfall amounts from the 16 members of the physical ensemble are shown in Table 5 and ranked according to REs. The member 5, 4 and 2 rank in the top 3 for event I (storm type 1) with relatively lower REs. For type 2 events, the member 4 and 12 show more stable performances, ranking in the top 5 for both event II and VI. For type 3 events, the member 5 and 7 are better choices with top 5 rankings for event IV and V. The top 4 members for event III (type 4) are member 4, 2, 16 and 3. It can be seen that the performances of the 16 members are quite distinct for different types of storm events. In addition, the difference among the 16 members varies a lot for a certain storm event. For example, the difference of REs for the member 8 (18.44%) and member 9 (-37.69%) reaches up to 56.13% for event I. While for event

V, the largest difference of RE among all the 16 members is only 9.10%. There are great uncertainties for the simulation of the different storm events using WRF model with different combinations of the physical parameterizations. It's hard to tell which parameterization combination is the best, but only to find the one with the best general performance. In this study, member 4 could be the best choice considering its stable top-rankings for storm type 1, 2 and 4, while member 9, 10, 11 and

12 have worse performance for storm type 1 and 4. For type 3 event, member 5 and 7 are better choices. However, in real-time rainfall prediction, there is a necessity to use physical ensemble since it is always tricky to tell the exact characteristics of the future storm before it happens, and the use of a determined combination of parameterizations which performs generally well cannot always lead to the best results. According to Table 5, the four members without cumulus parameterization have quite different performance for different event. For example, member 15 performs the best for event

IV, nevertheless, it performs the worst for event V. Comparing with the members containing cumulus parameterization, member 13, 14, 15 and 16 have no significant advantages or significant disadvantages for rainfall simulation. Taking event I as an example, the best one (member 16) in the four members without cumulus parameterization ranks fourth in the 16 members, whereas the worst one (member 13) ranks twelfth. However, few members without cumulus parameterization rank in the top 4, which means that it is necessary to use cumulus parameterization for the simulation of rainfall accumulation.

**[Table 5]**

In order to measure the magnitude of error for different storm types, all the REs use absolute values in the following analysis to calculate the average relative error (ARE) of the 16 members of the physical ensemble. The AREs of the 16 members for

the four storm types are shown in Table 6. It's interesting to note that the ranking of the model performance is type 1 > type 2 > type 3 > type 4, from the best to the worst. It means that the WRF model performs best for the storm event with even spatiotemporal distribution, while the type of storm event with highly uneven spatiotemporal distribution is hard for WRF to handle. The cumulative curves of the simulated and observed rainfall for the 6 storm events are shown in Fig. 3. Except for event I, the cumulative curves of the members are all below the observed ones for the other storm events. The shapes of 16

simulated cumulative curves are consistent with the observed ones for event I, II and VI (type 1 and type 2 events), indicating that the simulated rainfall occurrences always keep step with the observations. While for event IV, V and III (type 3 and type 4 events), the simulated starting and ending times of the rainfall durations are quite different from the observations. It can be found that type 1 and type 2 events have even rainfall distributions in space, while the spatial rainfall is unevenly distributed in space for type 3 and type 4 events. It seems that storms with rainfall evenly distributed in space

tend to have better simulation results in the temporal patterns of rainfall accumulations.

**[Table 6 and Figure 3]**

## 4.2 Simulations of the spatial rainfall distributions

In order to compare the simulation results of the different storm types in detail, seven verification indices are first calculated to evaluate the simulated rainfall distributions in space. Figure 4 and 5 respectively shows the values of the categorical indices and continuous indices for the 6 storm events with the 16 members of the physical ensemble.

It can be seen in Fig. 4 that PODs of storm type 1 and 2 (event I, II and VI) are all above 0.70 for the 16 members, which means that the events with even distributions can be accurately simulated regarding the rainfall occurrences in space. For the other two storm types, event IV with relatively lower $C_v$ performs better than event V and III. However, PODs of the 16 members for type 4 event (event III) are all close to zero, indicating that the WRF model can hardly to capture the storm occurrence in space. Event I and IV have nearly perfect scores of FBI, which are close to 1.0. For event II, III and VI, WRF

tends to overestimate the rainfall occurrences; while for event V, the model tends to have underestimations. Storm type 1 has the lowest FARs and the values are all under 0.20 in the 16 members, which means the WRF model has little false alarm possibility in space. While storm type 4 (event III) fails to be regenerated by the model in space because of the highest FARs (near 1.0). Storm type 3 outperforms storm type 2 with relatively lower FARs. CSI can be considered as a comprehensive description of accuracy. Storm type 1 with the highest CSIs performs the best in all the 16 members, while CSIs of storm

type 4 are all close to 0 showing that the simulation results are unreliable. CSIs of the other two storm types have little difference as a whole, but the index values are a little bit higher for events with more evenly-distributed rainfall in space.

Figure 5 shows that the values of RMSE have great change in different members for a certain event. RMSE is always regarded as the key quantitative index to estimate errors. Storm event II with the lowest $C_v$ always has the lowest RMSE for the 16 members, which means that the WRF model performs the best for storm event II in simulating the spatial rainfall

distributions. Except for member 1 and 4, the event III has the highest RMSE, and the values of 8 members exceed 100%. For the other four events, there is little difference of RMSEs in the 16 members. The MBE index contains the directions of errors, but in Fig. 5 absolute values of MBE are used. Storm type 1 has the lowest MBEs in the 16 members, and the MBEs of storm type 3 and 4 are higher than storm type 2. The values of SD also show variations for a certain storm type in different members. As a whole, SD and RMSE have similar patterns for different types of storm events. From Fig. 4 and 5, it

can be easily found that few values of the indices for member 13, 14, 15 and 16 are out of the range of the values for other 12 members, which indicates that there are always some members performing better than the four members without cumulus parameterization. It is helpful to use appropriate cumulus parameterization for the simulation of the spatial rainfall distribution.

[Figure 4 and Figure 5]

The average values of the 16 members for all the 7 indices are calculated to quantitative analysis the performance of the WRF model in spatial dimension for the four storm types. As shown in Table 7, the value of POD for storm type 1 is higher

than storm type 3 and 4. In addition, the value of CSI for storm type 1 is the highest and the value of FAR is the lowest in the four storm types. The lower values of RMSE and MBE for storm type 1 also indicate that the WRF model performs well for storm type 1. The simulations of type 3 events are worse than type 2 events, showing lower POD and higher RMSE values, though the FARs of the type 2 events are a little higher than type 3 events. The lowest POD, CSI and the highest FAR, RMSE can be found with storm type 4, which indicates the WRF model can hardly capture this kind of storm accurately in space. Since the index of RMSE shows the actual magnitude of errors without cancelling out the positive and negative errors, a correlation analysis is further carried out between RMSE and the spatial evenness indicator $C_v$. It's interesting to find that RMSE and $C_v$ have a good linear relationship and the correlation coefficient of the linear regression ($R^2$) can reach up to 0.8899 (shown by Fig. 6). This means that the WRF simulation error increases with the increase of the spatial rainfall unevenness in the study sites.

**[Table 7 and Figure 6]**

### 4.3 Simulations of the temporal rainfall patterns

The seven indices are also calculated in the temporal dimension to evaluate the simulated rainfall patterns in time. The values are respectively shown in Fig. 7 and 8. In Fig. 7, PODs of storm type 1 and 2 are all above 0.70 and much higher than storm type 3 and 4 in the 16 members. It indicates that storm type 1 and 2 can be accurately simulated regarding the rainfall occurrence in the temporal dimension. While the WRF model fails with storm type 4, with all PODs of the 16 members close to 0. For FBI, the scores of event I and IV are nearly perfect, but the other four events show tendencies of overestimating the rainfall occurrences in time, especially event VI. The lowest FAR values are also found with storm type 1 with all the values less than 0.20 in the 16 members. Storm type 4 has the highest FARs which are close to 1.0 in some members. Based on the FAR index, the ranking of the WRF performance in simulating temporal rainfall occurrences is type 1 > type 3 > type 2 > type 4, from the best to the worst. In the 16 members, CSIs of storm type 1 are always the highest, while CSIs of storm type 4 are always the lowest. It should be mentioned that the CSI is 0 in member 7, 11, 14 and 15 in storm event III, indicating bad simulation of the temporal rainfall occurrences for this type 4 event.

In Fig. 8, type 1 event has the lowest RMSEs in the 16 members, but the values are nearly 100%. Type 4 event has the highest RMSEs which are all above 250%. The other two types of storm events also have high RMSE values between 100% and 180%. We can say that the WRF model cannot perform well in simulating the temporal rainfall patterns for all the storm types. Storm type 1 has the lowest MBEs, and the MBE values of storm type 3 and 4 are relatively higher than storm type 2 in most members. All SDs are above 100% in the 16 members for the six events, with the lowest values found with event II. From Fig. 7 and 8, the same as the conclusions in the spatial dimension, most values of the indices for member 13, 14, 15 and 16 are in the range of the values for other 12 members, which indicates that there are always some members performing

better than the four members without cumulus parameterization. It is also necessary to use cumulus parameterization for the simulation of the temporal rainfall distribution.

**[Figure 7 and Figure 8]**

The average ensemble values for the 7 indices are also calculated for evaluating the performance of WRF model in simulating the temporal rainfall patterns. The results are shown in Table 8. The values of POD and CSI for storm type 1 are the highest and the values of FAR and RMSE are the lowest in the four storm types, which indicate that the WRF model performs best for storm type 1. The model performs the worst for storm type 4 with the lowest POD and CSI and the highest

FAR and RMSE. In general, the simulation results of the temporal rainfall patterns are unsatisfactory for all the four storm types. The linear relationship between RMSE and the temporal $C_v$ is also significant and the correlation coefficient of linear regression ($R^2$) is 0.7524 (shown by Fig. 9). It indicates that the simulation error also increases with the increase of the rainfall unevenness in the temporal dimension.

**[Table 8 and Figure 9]**

**5 Discussion**

In this study, the performances of 16 WRF physical members are estimated firstly by AREs for cumulative rainfall amounts and then by a two-dimensional verification scheme for spatiotemporal rainfall distributions. According to the spatiotemporal

evenness, six storm events are classified into four storm types. Storm type 1 has two-dimensional evenness of the rainfall which is even in spatiotemporal distribution. The WRF model performs best for simulating this storm type, not only for the cumulative rainfall amounts but also for the spatiotemporal distributions. Storm type 2 is only even in space, and the simulation results from the WRF ensemble are better than storm type 3 and 4. But compared with type 1, the cumulative rainfall amounts of type 2 events are seriously underestimated. Storm type 3 and 4 are both uneven in spatiotemporal

distribution, and the unevenness is especially remarkable for type 4 event. The simulations of WRF model are unsatisfactory for the spatiotemporal patterns of the two storm types. The simulation results of type 4 event are the worst among the four storm types. Some of the members even miss the whole storm duration in space and time. It is interesting to find that the WRF model tends to underestimate the rainfall amounts except for storm type 1. With more events being investigated in the study sites, the general simulation errors of the WRF model can be learnt by statistical analysis, which can help to build a

correction model to further improve the rainfall products of the WRF model.

For rainfall forecast operation, it is hard to identify the storm type before the storm occurs. Therefore it is important to find out the physical parameterizations which generally perform well. According to the REs of the 16 members for the six storm

events showed in Table 5, the AREs of the six storm events for one certain member are calculated. It is interesting to find that members containing BMJ have relatively higher AREs, which are 52.49% (member 9), 48.30% (member 10), 48.35% (member 11) and 49.05% (member 12) respectively. The relative lower AREs (34.02~39.50%) can be found in members which contain KF. The members containing GD perform better than members with BMJ while worse than members with KF.

The range of the AREs is 42.32~44.53%. The members without cumulus parameterization also perform better than members with BMJ while worse than members with KF and the range of the AREs is 39.55~49.16%. That is to say, the cumulus parameterizations have significant effect on the performance of the WRF model and BMJ performs the worst in the three cumulus parameterizations. Janjić (2000) indicated that BMJ had the poor performance in accurately reproducing the range and the intensity of the low-level jet. The strong ability of BMJ in simulating the upward transportation of vapor always

results in underestimation of the rainfall amount. That is the main reason why BMJ is not a good choice in the study area. Additionally, it is necessary to use cumulus parameterization for the simulation of the rainfall accumulation and spatiotemporal rainfall distribution in the study area. However, the threshold of the horizontal resolution need to be further discussed for whether to use the cumulus parameterization.

The uncertainties of the rainfall processes affect the choice of the physical parameterizations in a certain area. It is necessary

to select the most appropriate physical parameterizations to design the physical ensemble for rainfall simulation and prediction. In this study, the 16 members of the physical ensemble are constituted from two microphysics, two PBLs and three cumulus parameterizations, which are proved to be appropriate and widely used in the neighboring areas of the study sites (Hong et al, 2006; Miao et al, 2011; Pan et al, 2014). With the development of the WRF model, more sophisticated and realistic physical parameterizations could be developed and should be tested in the study area.

The verification of the WRF model has always been recognized as a worthy issue to be explored. In this study, a verification method which can estimate the rainfall simulations in both the spatial and the temporal dimension. It is assumed that the observations from rain gauges are accurate and representative for the two study sites. However, it brings uncertainties to use point-based observations to evaluate grid-based simulations. More grid-based observational data should be involved to improve the reliability of evaluation, especially those from weather radar and remote sensing.

Ultimately, the main goal of rainfall forecasts is to obtain efficient flood forecasts. The peak flood, flood peak appearance time, flood process are all significantly influenced by the rainfall accumulations and the spatiotemporal distribution of the rainfall (Schellekens et al, 2011; Cane et al, 2013; Fan et al, 2015). Event V which occurred on 21 July 2012 has caused the greatest flood during the past 10 year in Jing-Jin-Ji (Beijing-Tianjin-Hebei) area and received widespread attention in China. The 24 h rainfall accumulation was 155.43 mm in Zijingguan catchment and the peak flow reached 2580 $m^3/s$ at the

catchment outlet. In such cases, accurate rainfall simulations and predictions can do great help to flood warning. However, to analyze the usefulness of the WRF simulations to flood warning, the rainfall-runoff transformation processes should be further considered. This will involve many uncertainties, such as the choice of the rainfall-runoff model, the data used for model calibration, and the involvement of a real-time updating scheme, etc., which also have considerable impact on the

accuracy of the flood forecasting results. The exploration of different parameterizations from the flood warning purposes is an important issue and worthy to be discussed in further study.

**6 Conclusion**

In this study, the FNL data from NCAR provide the initial and boundary conditions for the WRF model, which is used for rainfall simulation of six representative storm events with the duration of 24h in Fuping and Zijingguan catchments, locating in the south and the north reaches of Daqinghe Basin in semi-humid area of North China. Two microphysics, two PBLs and three cumulus parameterizations are selected to develop 16 members of the physical ensemble of the WRF model. Both the cumulative amount and the spatiotemporal patterns of the simulated rainfall are analysed and verified. The relative error is used to evaluate the 24 h accumulated areal rainfall. The spatial rainfall distributions and temporal rainfall patterns are verified by a two-dimensional verification scheme including 4 categorical and 3 continuous indices. The six storm events are classified into four types based on the spatiotemporal evenness of the rainfall. In general, the ranking of the average model performance for different storm types is type 1 > type 2 > type 3 > type 4, from the best to the worst regarding both the cumulative rainfall amounts and the spatiotemporal rainfall patterns. Negative correlation is found between the simulation error and the rainfall evenness in both spatial and temporal dimensions. Storm events with more evenly-distributed rainfall tend to have better simulation results in space and time. In addition, for the small catchment scale, accumulated areal rainfall is more important than the spatiotemporal rainfall distributions. According to the REs of rainfall accumulations, member 4 is the better choice for storm type 1, 2 and 4, while member 9, 10, 11 and 12 have the worse performance for storm type 1 and 4. For type 3 event, member 5 and 7 are the better choices. It provides a reference for choosing the optimal ensemble in the study area for different storm types.

This study provides a reference for ensemble simulation of different rainfall types in semi-humid area of China the WRF model. However, the simulated rainfall has relatively large errors and the simulation results of the temporal rainfall patterns are always unreliable, especially the results of event III and V which cannot be used directly in hydrological studies. Data assimilation has been proved to be an effective method in improving the rainfall simulation results of the WRF model by many studies (Ha and Lee, 2012; Liu et al, 2012; Routray et al., 2012). Data assimilation can ingest various sources of observations (surface observed data, radar data, satellite data and sounding data, etc.) into the WRF model products and then use the respective error statistics to update and correct the WRF model products (Wan and Xu, 2011; Ha et al., 2014; Xie et al., 2016). More studies should be carried out in the study sites with the assistance of data assimilation so that the rainfall products from WRF model can be further improved.

**Acknowledgements:** This study was supported by the National Natural Science Foundation of China (Grant No. 51409270), the National Key Research and Development Project (Grant No. 2016YFA0601503), the International Science and Technology Cooperation Program of China (Grant No. 2013DFG70990), the Foundation of China Institute of Water

Resources and Hydropower Research (1232) and the Open Research Fund Program of State Key Laboratory of Hydrology-Water Resources and Hydraulic Engineering (2014490611).

**Author Contributions:** All the authors have contributed to the conception and development of this manuscript. Jiyang Tian carried out the analysis and wrote the paper. Jia Liu and Fuliang Yu conceived and designed the framework. Chuanzhe Li provided assistance in calculations and figure productions.

**Competing interests:** The authors declare that they have no conflict of interest.

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

# Figure captions

**Figure 1: The nested domains and the orography of the Fuping catchment and Zijingguan catchment.**

**Figure 2**: The location of Daqinghe catchment in the Northern China (light shading) and the locations of the two study sites in Daqinghe catchment.

**Figure 3**: Cumulative curves of the observed and simulated areal rainfall for the six storm events.

**Figure 4**: Spatial values of the four categorical indices for different storm events with the 16 members of the physical ensemble.

**Figure 5**: Spatial values of the three continuous indices for different storm events with the 16 members of the physical ensemble.

**Figure 6**: The relationship between RMSE and Cv in the spatial dimension.

**Figure 7**: Temporal values of the four categorical indices for different storm events with the 16 members of the physical ensemble.

**Figure 8**: Temporal values of the three continuous indices for different storm events with the 16 members of the physical ensemble.

**Figure 9**: The relationship between RMSE and Cv in the temporal dimension.

# Table captions

Table 2: The constitution of the WRF physical ensembles.

Table 2: Durations and rainfall accumulations of the six selected 24-h storm events.

Table 3: Spatial and temporal Cv of the observed rainfall for the six storm events.

Table 4: Rain/ no rain contingency table for the WRF simulation against observation.

Table 5: Rankings of the 16 members of the physical ensemble according to RE (%) of the simulated rainfall accumulations for the storm events.

Table 6: AREs of the 16 members of the physical ensemble for the four types of storm events (%).

Table 7: Average index values of the 16 members of the physical ensemble for the simulations of the spatial rainfall distributions.

Table 8: Average index values of the 16 members of the physical ensemble for the simulations of the temporal rainfall patterns.

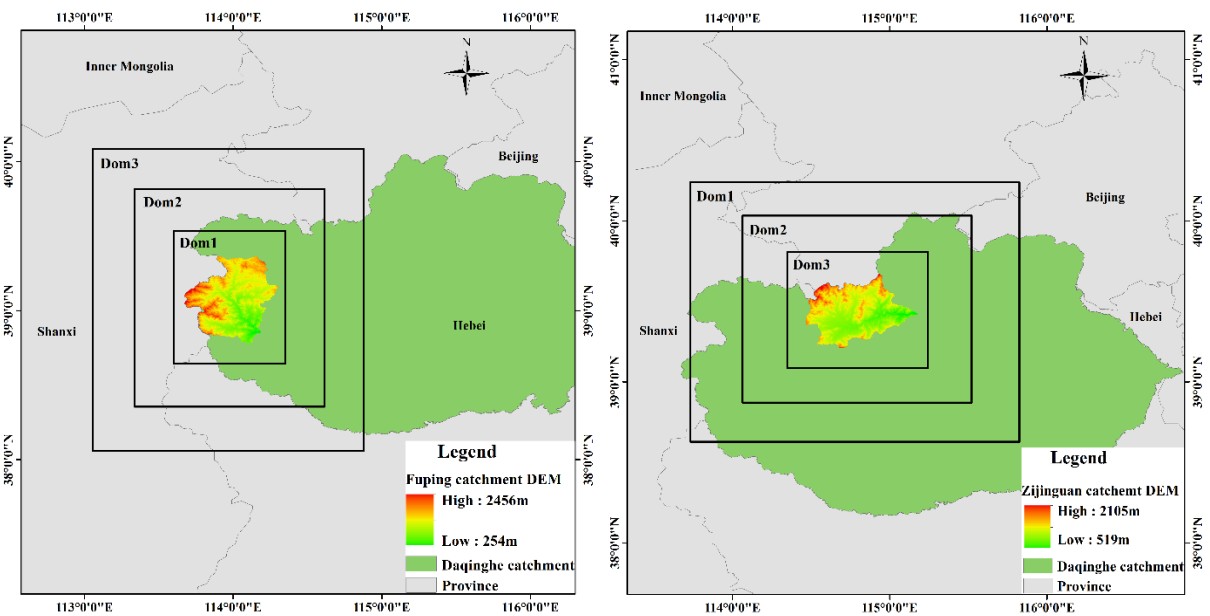

**Figure 1.** The nested domains and the orography of the Fuping catchment and Zijingguan catchment

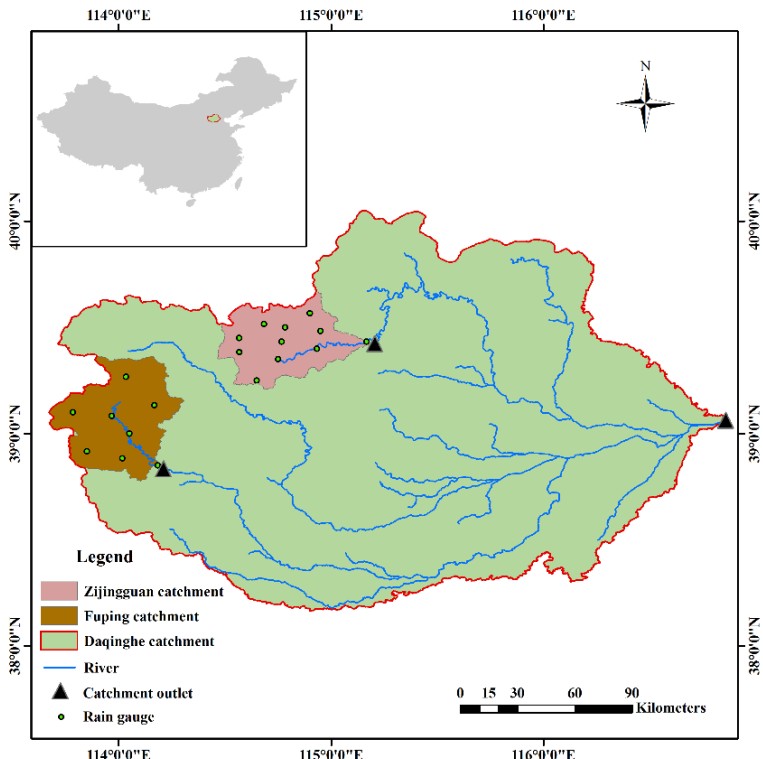

**Figure 2.** The location of Daqinghe catchment in the Northern China (light shading) and the locations of the two study sites in Daqinghe catchment

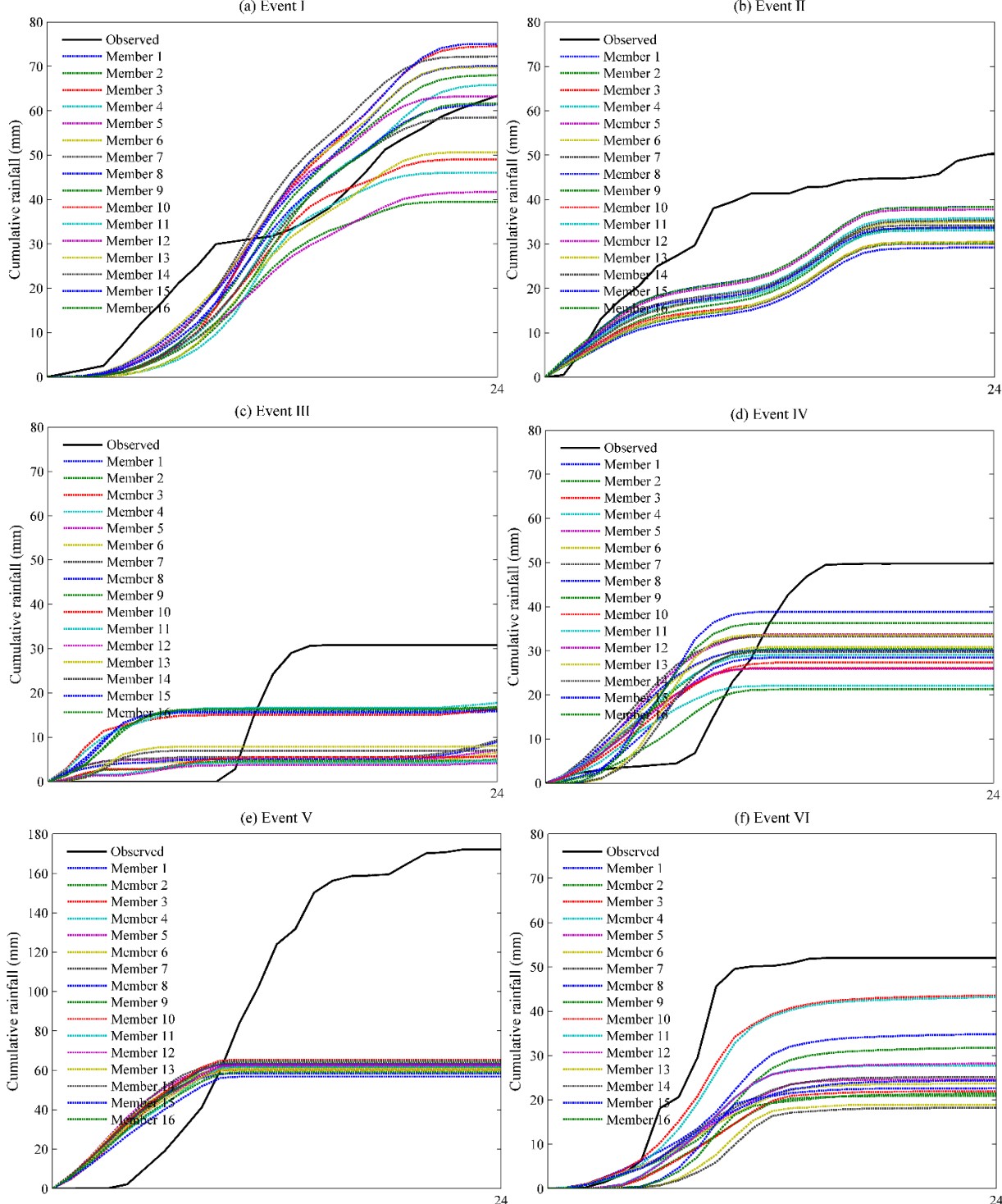

**Figure 3.** Cumulative curves of the observed and simulated areal rainfall for the six storm events

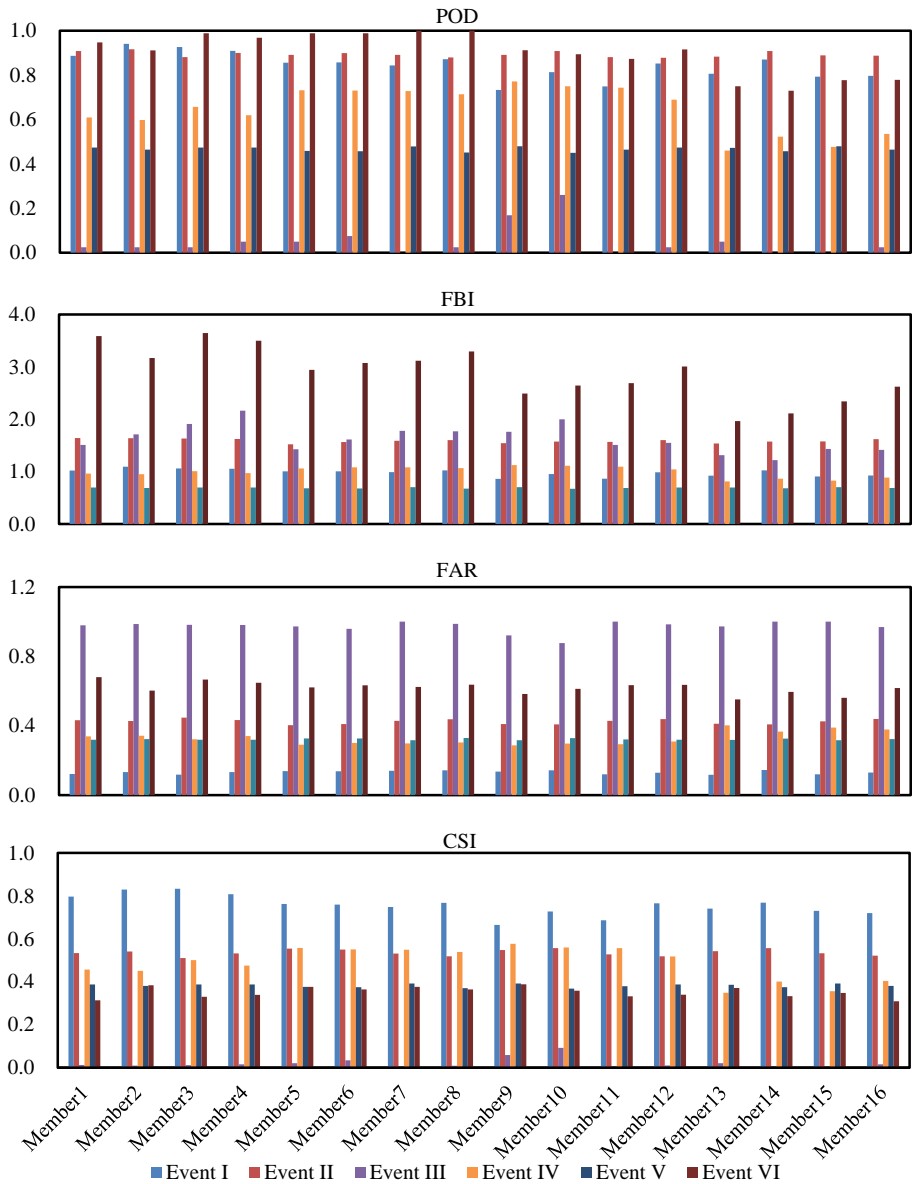

5    **Figure 4.** Spatial values of the four categorical indices for different storm events with the 16 members of the physical ensemble

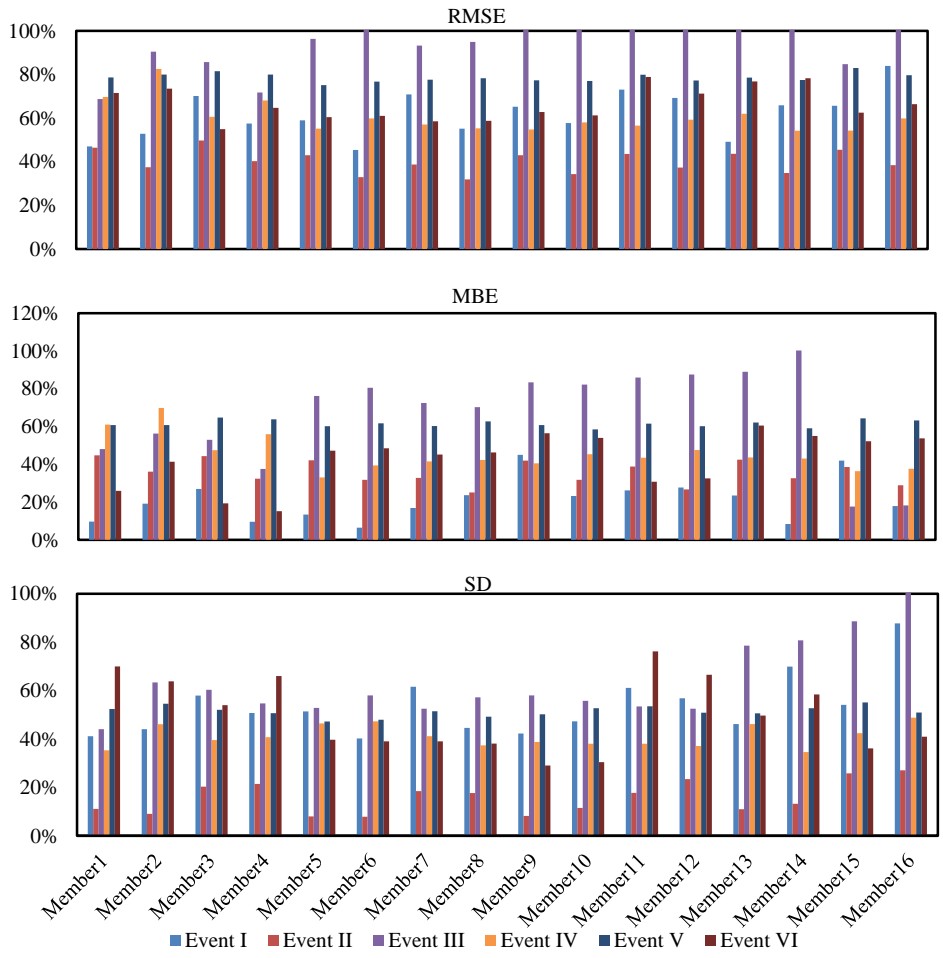

**Figure 5.** Spatial values of the three continuous indices for different storm events with the 16 members of the physical ensemble

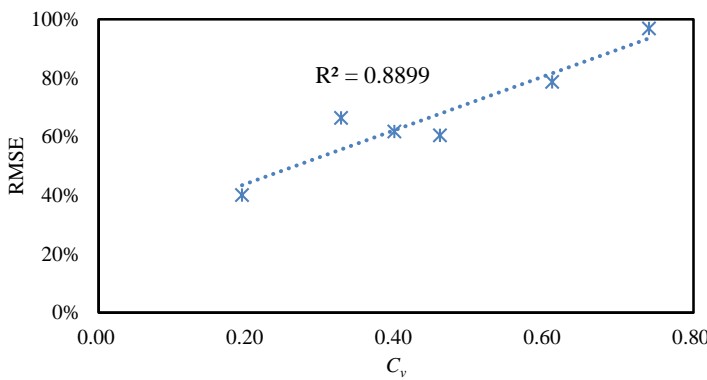

**Figure 6.** The relationship between RMSE and $C_v$ in the spatial dimension

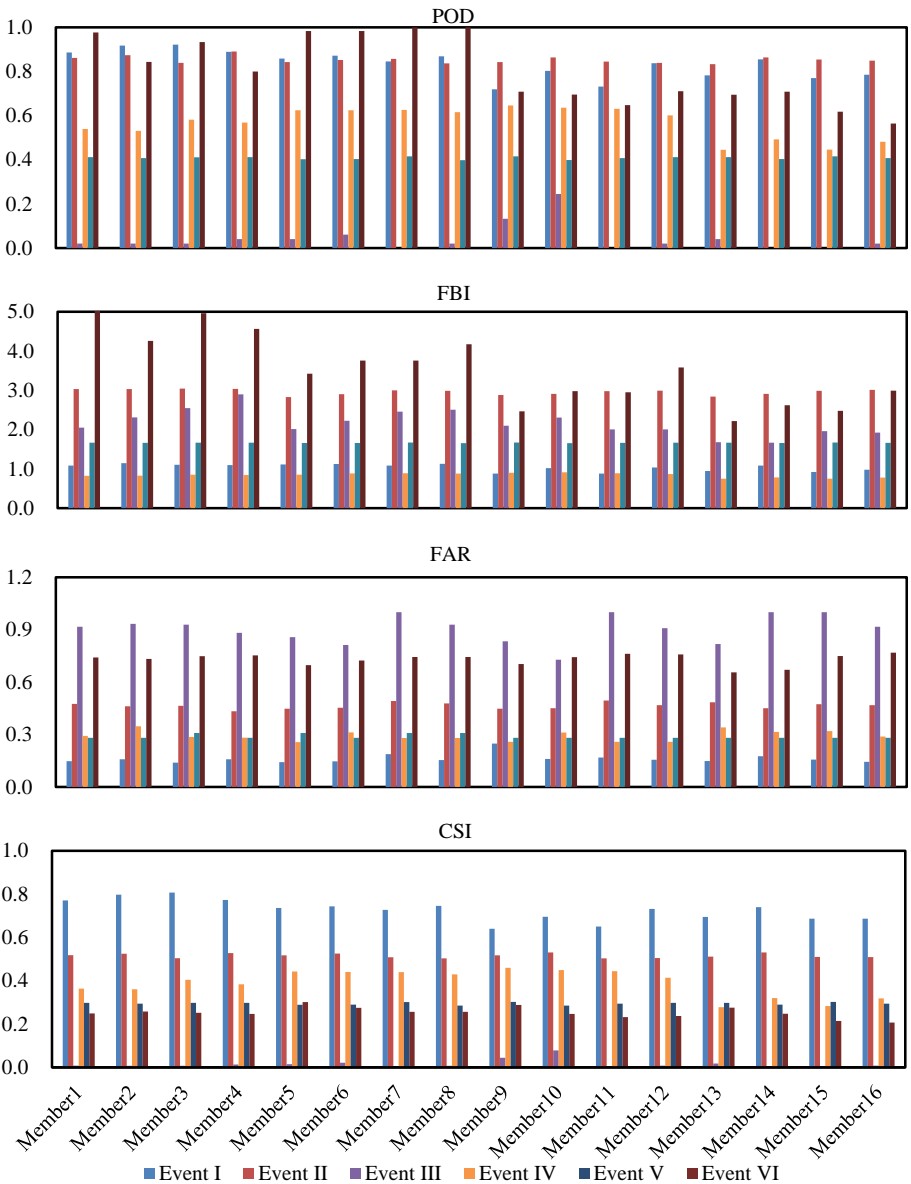

5    **Figure 7**. Temporal values of the four categorical indices for different storm events with the 16 members of the physical ensemble

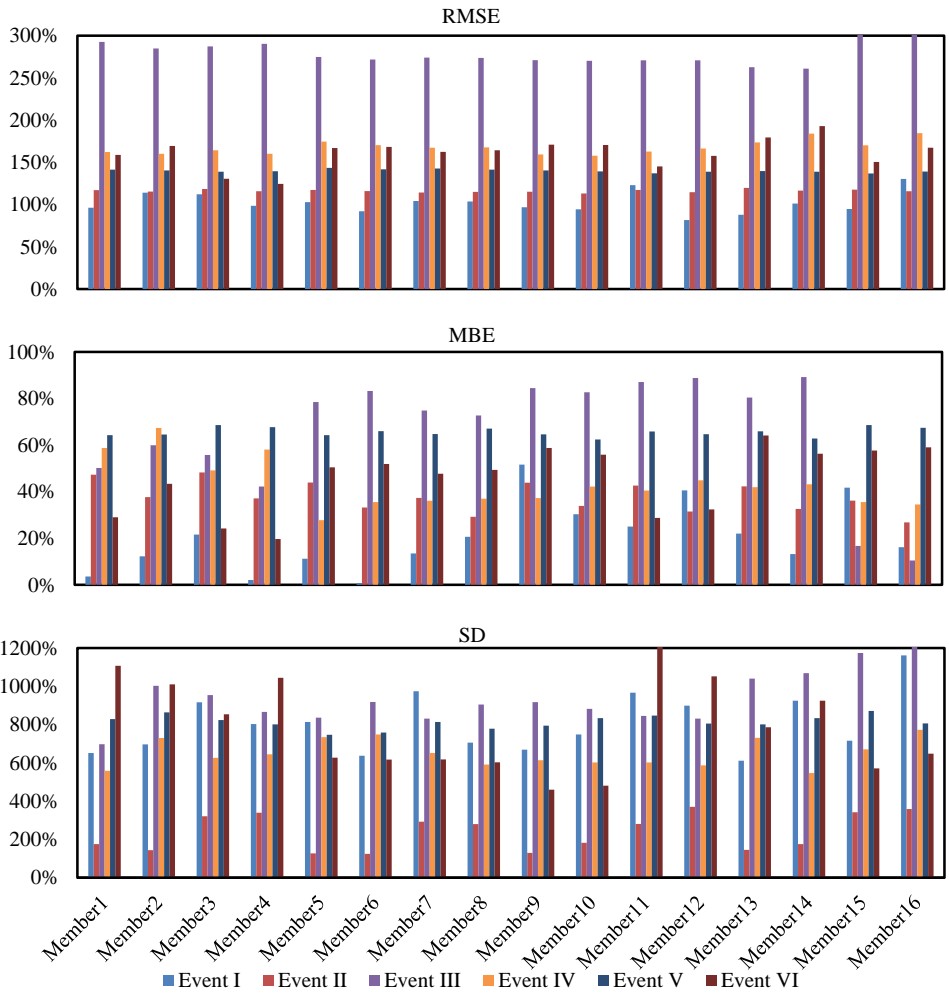

**Figure 8.** Temporal values of the three continuous indices for different storm events with the 16 members of the physical ensemble

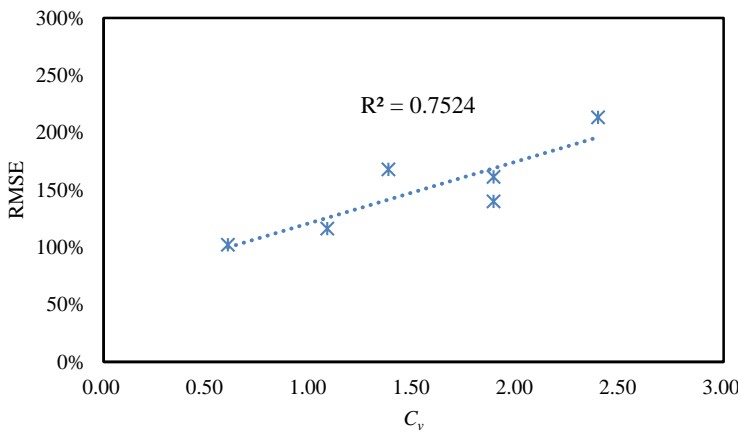

**Figure 9.** The relationship between RMSE and $C_v$ in the temporal dimension

**Table 1.** The constitution of the WRF physical ensemble

| Ensemble ID | Microphysics | PBL | Cumulus parameterization |
|:---:|:---:|:---:|:---:|
| 1 | Lin | YSU | KF |
| 2 | WSM6 | YSU | KF |
| 3 | Lin | MYJ | KF |
| 4 | WSM6 | MYJ | KF |
| 5 | Lin | YSU | GD |
| 6 | WSM6 | YSU | GD |
| 7 | Lin | MYJ | GD |
| 8 | WSM6 | MYJ | GD |
| 9 | Lin | YSU | BMJ |
| 10 | WSM6 | YSU | BMJ |
| 11 | Lin | MYJ | BMJ |
| 12 | WSM6 | MYJ | BMJ |
| 13 | Lin | YSU | / |
| 14 | WSM6 | YSU | / |
| 15 | Lin | MYJ | / |
| 16 | WSM6 | MYJ | / |

**Table 2.** Durations and rainfall accumulations of the six selected 24-h storm events

| Event ID | Catchment | Storm start time | Storm end time | Accumulated 24-h rainfall (mm) |
|----------|-----------|------------------|----------------|-------------------------------|
| I | Fuping | 29/07/2007 20:00 | 30/07/2007 20:00 | 63.38 |
| II | Fuping | 30/07/2012 10:00 | 31/07/2012 10:00 | 50.48 |
| III | Fuping | 11/08/2013 07:00 | 12/08/2013 07:00 | 30.82 |
| IV | Zijingguan | 10/08/2008 00:00 | 2008/08/10 24:00 | 45.53 |
| V | Zijingguan | 21/07/2012 04:00 | 22/07/2012 04:00 | 155.43 |
| VI | Zijingguan | 06/06/2013 22:00 | 07/06/2013 22:00 | 52.06 |

**Table 3.** Spatial and temporal $C_v$ of the observed rainfall for the six storm events

| Indices | Type 1 | Type 2 | | Type 3 | | Type 4 |
|---|---|---|---|---|---|---|
| | Event I | Event II | Event VI | Event IV | Event V | Event III |
| Spatial $C_v$ | 0.3975 | 0.1927 | 0.3258 | 0.4588 | 0.6098 | 0.7400 |
| Temporal $C_v$ | 0.6011 | 1.0823 | 1.8865 | 1.3779 | 1.8865 | 2.3925 |

**Table 4.** Rain/ no rain contingency table for the WRF simulation against observation

| WRF/observations | Rain | No rain |
|---|---|---|
| Rain | $NA$ (hit) | $NB$ (false alarm) |
| No rain | $NC$ (failure) | $ND$ (correct negative) |

**Table 5.** Rankings of the 16 members of the physical ensemble according to RE (%) of the simulated rainfall accumulations for the storm events

| Ranking | Type 1 | Type 2 | | Type 3 | | Type 4 |
|---|---|---|---|---|---|---|
| | I | II | VI | IV | V | III |
| 1 | Member 5 (-0.17) | Member 8 (-24.05) | Member 3 (-16.32) | Member 15 (-21.89) | Member 10 (-57.89) | Member 4 (-42.41) |
| 2 | Member 4 (3.85) | Member 12 (-25.12) | Member 4 (-17.03) | Member 5 (-25.77) | Member 2 (-58.91) | Member 2 (-45.35) |
| 3 | Member 2 (7.23) | Member 4 (-29.12) | Member 1 (-33.05) | Member 7 (-27.03) | Member 7 (-59.22) | Member 16 (-46.55) |
| 4 | Member 16 (-7.47) | Member 10 (-30.09) | Member 2 (-38.87) | Member 16 (-27.13) | Member 1 (-59.31) | Member 3 (-46.93) |
| 5 | Member 6 (10.17) | Member 6 (-30.72) | Member 12 (-45.79) | Member 6 (-32.19) | Member 5 (-59.54) | Member 15 (-47.59) |
| 6 | Member 1 (10.55) | Member 14 (-32.10) | Member 11 (-46.60) | Member 13 (-32.43) | Member 12 (-59.57) | Member 1 (-48.59) |
| 7 | Member 15 (-10.99) | Member 7 (-32.23) | Member 7 (-51.66) | Member 9 (-33.17) | Member 4 (-60.15) | Member 7 (-69.79) |
| 8 | Member 14 (-10.83) | Member 2 (-33.27) | Member 5 (-52.76) | Member 8 (-33.90) | Member 11 (-60.20) | Member 8 (-70.95) |
| 9 | Member 7 (13.96) | Member 15 (-33.36) | Member 8 (-53.12) | Member 11 (-36.23) | Member 9 (-60.24) | Member 13 (-73.88) |
| 10 | Member 3 (17.54) | Member 16 (-34.03) | Member 6 (-54.57) | Member 1 (-37.53) | Member 6 (-60.81) | Member 14 (-77.06) |
| 11 | Member 8 (18.44) | Member 11 (-34.59) | Member 15 (-56.48) | Member 10 (-39.93) | Member 3 (-61.17) | Member 5 (-77.19) |
| 12 | Member 13 (-20.12) | Member 3 (-39.71) | Member 10 (-57.85) | Member 14 (-40.24) | Member 14 (-62.37) | Member 6 (-78.70) |
| 13 | Member 10 (-22.63) | Member 13 (-39.72) | Member 16 (-58.78) | Member 3 (-42.64) | Member 8 (-62.43) | Member 10 (-81.42) |
| 14 | Member 11 (-27.30) | Member 9 (-40.24) | Member 9 (-59.85) | Member 12 (-42.99) | Member 13 (-65.12) | Member 9 (-83.77) |
| 15 | Member 12 (-34.24) | Member 5 (-40.41) | Member 13 (-63.66) | Member 4 (-51.58) | Member 16 (-65.73) | Member 11 (-85.16) |
| 16 | Member 9 (-37.69) | Member 1 (-42.15) | Member 14 (-65.04) | Member 2 (-53.36) | Member 15 (-66.99) | Member 12 (-86.59) |

**Table 6.** AREs of the 16 members of the physical ensemble for the four types of storm events (%)

| Type 1 | Type 2 | | Type 3 | | Type 4 |
|--------|--------|--------|--------|--------|--------|
| I | II | VI | IV | V | III |
| 15.82 | 33.80 | 43.96 | 48.22 | 64.18 | 66.37 |

**Table 7.** Average index values of the 16 members of the physical ensemble for the simulations of the spatial rainfall distributions

| Types of storm events | | Categorical indices | | | | Continuous indices (%) | | |
|---|---|---|---|---|---|---|---|---|
| | | POD | FBI | FAR | CSI | RMSE | MBE | SD |
| Type 1 | Event I | 0.8440 | 0.9815 | 0.1313 | 0.7565 | 61.74 | 21.21 | 53.54 |
| Type 2 | Event II | 0.8934 | 1.5877 | 0.4238 | 0.5357 | 40.07 | 35.67 | 15.74 |
| | Event VI | 0.9014 | 2.8866 | 0.6187 | 0.3516 | 66.36 | 42.74 | 49.78 |
| Type 3 | Event IV | 0.6460 | 0.9974 | 0.3285 | 0.4873 | 60.46 | 45.49 | 41.10 |
| | Event V | 0.4671 | 0.6906 | 0.3215 | 0.3821 | 78.65 | 61.51 | 51.36 |
| Type 4 | Event III | 0.0503 | 1.6301 | 0.9731 | 0.0194 | 96.88 | 66.14 | 63.53 |

**Table 8.** Average index values of the 16 members of the physical ensemble for the simulations of the temporal rainfall patterns

| Types of storm events | | Categorical indices | | | | Continuous indices (%) | | |
|---|---|---|---|---|---|---|---|---|
| | | POD | FBI | FAR | CSI | RMSE | MBE | SD |
| Type 1 | Event I | 0.8341 | 1.0389 | 0.1621 | 0.7264 | 102.18 | -20.37 | 805.67 |
| Type 2 | Event II | 0.8531 | 2.9596 | 0.4654 | 0.5153 | 116.27 | -37.74 | 236.57 |
| | Event VI | 0.8044 | 3.5119 | 0.7310 | 0.2527 | 161.29 | -45.55 | 787.85 |
| Type 3 | Event IV | 0.5683 | 0.8429 | 0.2931 | 0.3894 | 167.89 | -43.11 | 650.35 |
| | Event V | 0.4083 | 1.6646 | 0.2880 | 0.2947 | 140.00 | -65.60 | 812.78 |
| Type 4 | Event III | 0.0427 | 2.1653 | 0.9040 | 0.0148 | 253.27 | -66.08 | 948.23 |