# Peer review of "Numerical rainfall simulation with different spatial and temporal evenness by using WRF multi-physics ensemble"

_Natural Hazards and Earth System Sciences, 2016_

## Short Comment (SC1) · 24 Dec 2016

The manuscript contains an interesting comparison of various ensembles of physical parameterizations on precipitation forecasts of WRF model for several catchments in Northern China. It is constructive and some comments are described below: 1. Since each storm is 24 hours, it is assumed the authors initialize the model at the start of the storm. It would be informative to indicate when the model was initialized. This would also let the readers know how any approaches to spin-up (if any) was dealt with. 2. It is important to let the readers know the timescale for integration step of WRF model. 3. It seems that the model may produce some insignificant precipitation (less than 0.1 mm/hr). Could you explain it? 4. Why do you choose 40 vertical layers with 1 km

horizontal resolution? Could you give some details on the number of vertical layers used in the study? 5. If it is possible, more data would be better to firm the conculsion.

---

## Author Comment (AC1) · 25 Dec 2016

We thanks for the comments from Professor Zhang and the comments may help to improve the readability of the paper. The explaination to the comments is showed below point-by-point: 1.Since each storm is 24 hours, it is assumed the authors initialize the model at the start of the storm. It would be informative to indicate when the model was initialized. This would also let the readers know how any approaches to spin-up (if any) was dealt with. Reply: In this paper, the start time of the storm can be seen in Table 2. Due to a spin-up period of 6 hours is needed to develop the smaller scale convective features, the start of the model integration is 6 h earlier than the storm start time and the end time of the model integration is consistent with the storm end time. 2.It is important

to let the readers know the timescale for integration step of WRF model. Reply: The WRF developers recommend a timestep in seconds of 6×dx (in km), where dx is the grid spacing. The integration step of WRF used in this study follows exactly the '6×dx' rule and the integration step is 6s for innermost domain. 3.It seems that the model may produce some insignificant precipitation (less than 0.1 mm/hr). Could you explain it? Reply:As Professor Zhang mentioned, the threshold is 0.1mm/hr and the insignificant precipitation (less than 0.1 mm/h) is regarded as no rain. 4.Why do you choose 40 vertical layers with 1 km horizontal resolution? Could you give some details on the number of vertical layers used in the study? Reply:It is a good comment. The vertical layers between 25 and 55 are commonly used in WRF model. I think Professor Zhang may consider that the vertical layers may have effect on the performance of WRF model. However, it needs to do a lot of experiment to find the optimal number of layers. Qie et al. (2014) simulated the storm event occurred in Beijing, which is near the study area in the manuscript. The inner domain is 2-km and the vertical layers are set to be only 27. Aligo et al. (2009) indicated that the QPF forecasts cannot be always improved by adding the vertical layers with 4-km horizontal resolution in American Midwest. In my opinion, it is an interesting issue to investigate the combinations of the vertical layers and the horizontal resolution, but this is not the main concern of this study. We hope to obtain meaningful conclusions in further study. The two references are followed: Aligo E.A., Gallus W.A., Segal M., 2009. On the impact of WRF model vertical grid resolution on Midwest summer rainfall forecasts. Weather Forecast. 24, 575-594. Qie X., Zhu R., Yuan T., Wu X., Li W., Liu D., 2014. Application of total-lightning data assimilation in a mesoscale convective system based on the WRF model. Atmos. Res. s145–146, 255-266. 5.If it is possible, more data would be better to firm the conclusion. Reply: The conclusion of the study can provide a reference for the ensemble rainfall simulation in semi-humid and semi-arid areas of China. As the paper mentioned in section 6 "conclusion", more strom events should be investigated and simulated. More studies will be carried out in the study sites for further research.

---

## Referee Comment (RC1) · Anonymous Referee #1 · 4 Jan 2017

This paper try to evaluate a multi-physics ensemble of simulations for six cases at two different regions. The model runs are performed with a spatial resolution of 1km in the inner domain. The physical parametrizations tested are two microphysics schemes, two PBL schemes and 3 Cumulus schemes. Cumulus parametrizations are not needed at this spatial resolution (1km,3km ,and even 9km). Therefore, at this point the full paper makes no sense for this reviewer.

In addition I provide some more comments.

- The nomenclature is not right. The authors call ensemble N, to the member N of the ensemble.

- Plots showing the domains and orography would be desirable.

- What are the differences between semi-humid and semi-arid? The authors state that in both regions annual precipitation is 600mm.

-Fig 1, does not show the location of the rain gauges.

-What is the criterion to choose 0.4 as critical value for Cv?

- The way of calculating mean observed precipitation could be not appropriated, specially for "uneven" cases. Why not using directly the output of the model at the observation sites?

-The way of presenting the results should be improved. Figures 6 and 7 do not permit to extract fast conclusions.

---

## Referee Comment (RC2) · Anonymous Referee #2 · 5 Jan 2017

This paper describes a study on using WRF to downscale six rainfall events in North China for flood warning purposes. The authors have explored the impacts of different parameterizations on the simulation results. The work has been quite thorough with meticulous details. As WRF is increasingly applied in the natural hazard field, this paper is within the remit of the journal and its content is of interest to the readers. However, there are several issues that should be addressed for the paper to be acceptable.

1) please explain why a 6 hour spin-up period is used (e.g., why not 12 hours or other times);

2) 'The critical value of Cv is 0.4 and 1.0 for evaluating the rainfall evenness ' Please

explain how they are derived;

3) at fine spatial simulation resolutions (as pointed out by Referee #1), WRF is effectively running at CPM mode (Convection-permitting model) in which the dynamics of atmospheric convection is treated with sufficient accuracy in order to make it viable to switch off convection parametrization. It would be interesting to run your WRF model again without Cumulus parameterization and compare the results with the Cumulus parameterizations.

4) it would be helpful to know if any of the six rainfall events have caused any floods in the two study sites. Please explain which WRF simulations are useful to the flood warning purposes, and which are not (ultimately, this is the main goal of WRF applications). Do different parameterizations make any differences for warning purposes?

5) Language issues: The paper has several typos/grammatical errors. Please go through the whole paper carefully to remove them. For example, ' the precipitation easily cause flood...(causes)', ' . . . which is trick for forecasting accurately (tricky)', ' found by a lot of simulation. . . (simulations) ', etc.

---

## Referee Comment (RC3) · Anonymous Referee #2 · 9 Jan 2017

I am pleased with the answers so far and look forward to the updated manuscript.

---

## Author Comment (AC2) · 9 Jan 2017

Point 1: Cumulus parameterizations are not needed at this spatial resolution (1km, 3km, and even 9km). Therefore, at this point the full paper makes no sense for this reviewer.

Reply: We appreciate the referee for raising the issue which is worthy to be discussed. In general, WRF users tend to have fixed mindset that the cumulus parameterisations (CPs) are invalid with fine horizontal resolutions. As told by the WRF developers, the model is run at CPM mode where the dynamics of atmospheric convection is already treated with sufficient accuracy. This is also pointed out by the anonymous referee #2. However, in reality, the selection of different CPs at fine resolutions does make substantial differences in numerical rainfall simulation. This can be proved by the results of our study: the use of different CPs with identical settings of the other parameterisations resulted in different simulated rainfall. Many other studies also indicate that CPs have significant effect on the performance of the WRF model at fine resolutions, especially the three CPs (KF, GD and BMJ) discussed in our study (Argüeso et al, 2011; Evans et al, 2012; Pei et al, 2014). Among them, Shepherd et al (2016) explored the sensitivity of hurricane track to four cumulus parameterizations, including KF, BMJ, G-3 and TD, with the nested domains 1.33km, 4km and 12km. Madala et al (2013 and 2014) evaluated the performance of KF, GD and BMJ for the simulation of thunderstorm events, with the resolutions of the three nested domains being 3km, 9km and 27km. Remesan et al (2015) studied the WRF model sensitivity to the choice of parameterisations: 4 nested domains (1km, 3km, 9km and 27km) are used and the cumulus parameterisations of GD, BMJ, KF1 and KF2 are investigated. We do not approve that Referee #1 think the study is "no sense" based only on the unapproved point of CPs. The WRF model is still developing and explorations of the model users help break many boundaries. Like the widely accepted "6h spin-up time", the "1:3 downscaling ratio", and the "6-grid buffering zone", they are all proved not to be unchangeable rules. In order to make the manuscript more convincing, as suggested by Referee #2, we would like to compare the current rainfall simulation results with additional runs by masking CPs. This will be done by adding 4 members of the ensemble in Table 1 (Member 13-16). The references mentioned in the reply are provided as follows: Argüeso D, Hidalgomuñoz J M, Gámizfortis S R, Estebanparra M J, Dudhia J, Castrodiez Y. Evaluation of WRF parameterizations for climate studies over Southern Spain using a multistep regionalization. Journal of Climate, 2011, 24(21):5633-5651. Evans J P, Ekström M, Ji F. Evaluating the performance of a WRF physics ensemble over South-East Australia. Climate Dynamics, 2012, 39(6):1241-1258. Pei L, Moore N, Zhong S, Luo L, Hyndman D W, Heilman W E, Gao Z. WRF Model sensitivity to land surface model and cumulus parameterization under short-term climate extremes over the Southern Great Plains of the United States. Journal of Climate, 2014, 27(20):7703-7724. Shepherd T J, Walsh K J. Sensitivity of hurricane track to cumulus parameterization schemes in the WRF model for three intense tropical cyclones: impact of convective asymmetry. Meteorology and Atmospheric Physics, 2016:1-30. Madala S, Satyanarayana A N V, Tyagi B. Performance evaluation of convective parameterization schemes of WRF-ARW Model in the simulation of premonsoon thunderstorm events over Kharagpur using STORM Data Sets. International Journal of Computer Applications, 2013, 17(15):43-50. Madala S, Satyanarayana A N V, Rao T N. Performance evaluation of PBL and cumulus parameterization schemes of WRF ARW model in simulating severe thunderstorm events over Gadanki MST radar facility - Case study. Atmospheric Research, 2014, 139(6):1-17. Remesan R, Bellerby T, Holman I, Frostick L. WRF model sensitivity to choice of parameterization: a study of the 'York Flood 1999'. Theoretical and Applied Climatology, 2015, 122(1):229-247.

Point 2: The nomenclature is not right. The authors call ensemble N, to the member N of the ensemble.

Reply: Thanks for the suggestion and they will be revised accordingly.

Point 3: Plots showing the domains and orography would be desirable.

Reply: The following figures will be added to show the nested domains and the orography of the two study catchments.

Point 4: What are the differences between semi-humid and semi-arid? The authors state that in both regions annual precipitation is 600mm.

Reply: The dividing line between semi-humid and semi-arid region in China is an annual precipitation of 400mm. The two catchments in this study are located in semi-humid climatic region. The Haihe River basin which embraces the two study catchments covers a vast area of Northern China with both semi-humid and semi-arid climatic conditions. That is why the term "semi-humid and semi-arid" is used. In the revision, it will be clarified clearly.

Point 5: Fig 1, does not show the location of the rain gauges.

Reply: The locations of rain gauges will be added in Fig 1.

Point 6: What is the criterion to choose 0.4 as critical value for Cv?

Reply: This is a good point and readers may also have the same query while reading the paper. In order to learn the spatial and temporal evenness of the rainfall in the two study catchment, both spatial and temporal Cv of the storm events from 1985 to 2015 are all calculated. In reality, rainfall in Northern China is much more uneven than the south and it is impossible to find absolute even rainfall in both space and time. So we chose a threshold of 5%, which is also considered in other statistical analyses in the same area, as the critical value to separate even and uneven rainfall events. With the threshold, we found the two critical values of 0.4 for the spatial Cv and 0.6 for the temporal Cv. That is to say, the storm events with the spatial Cv below 0.4 or with the temporal Cv below 1.0 account for 5% of the total storm events from 1985 to 2015 in the study area. Explanations will be added in the manuscript to clarify this issue.

Point 7: The way of calculating mean observed precipitation could be not appropriated, specially for "uneven" cases. Why not using directly the output of the model at the observation sites?

Reply: In the manuscript, the simulated rainfall from the WRF model is evaluated from three aspects, i.e., the 24h areal rainfall accumulation, the spatial rainfall distribution and the temporal rainfall distribution. Firstly, the mean observed rainfall is used to calculate the accumulated areal rainfall, which is treated as an overall quantity metric. Secondly, the spatial and temporal distributions of the simulated rainfall are evaluated by a two-dimensional verification scheme comprised of 4 categorical indices and 3 continuous indices. At this stage of verification, the model outputs at the observation sites are directly used. In the spatial dimension, the observed and simulated rainfall accumulations at each rain gauge are compared; and in the temporal dimension, the comparison is made on the areal rainfall at each time step. Detailed explanations are
provided in the methodology part of Section 3.2.

Point 8: The way of presenting the results should be improved. Figures 6 and 7 do not permit to extract fast conclusions.

Reply: The storm types will be indicated by the members of ensemble, which can help readers find the corresponding event in Figure 3, 4, 6 and 7. If the referee has better suggestion to improve the way of presenting the results, we are pleased to accept it in detail.

[Figure: Map showing catchment area with legend]

**Fig. 1.** Figure1

[Figure]

[Figure]

**Fig. 2.** Domains and orography

**Table 1.** The constitution of the WRF physical ensemble

| Member ID | Microphysics | PBL | Cumulus parameterisation |
|---|---|---|---|
| 1 | Lin | YSU | KF |
| 2 | WSM6 | YSU | KF |
| 3 | Lin | MYJ | KF |
| 4 | WSM6 | MYJ | KF |
| 5 | Lin | YSU | GD |
| 6 | WSM6 | YSU | GD |
| 7 | Lin | MYJ | GD |
| 8 | WSM6 | MYJ | GD |
| 9 | Lin | YSU | BMJ |
| 10 | WSM6 | YSU | BMJ |
| 11 | Lin | MYJ | BMJ |
| 12 | WSM6 | MYJ | BMJ |
| *13* | *Lin* | *YSU* | / |
| *14* | *WSM6* | *YSU* | / |
| *15* | *Lin* | *MYJ* | / |
| *16* | *WSM6* | *MYJ* | / |

**Fig. 3.** Table1

---

## Author Comment (AC3) · 9 Jan 2017

Point 1: please explain why a 6 hour spin-up period is used (e.g., why not 12 hours or other times).

Reply: The spin-up period is necessary for WRF model and the widely used lengths are 6 hours (Givati et al, 2012), 12 hours (Hu et al, 2010) and 24 hours (Wang et al, 2012). Before we decide to use the 6h spin-up period, longer spin-up times were also tried for the six storm events. Results did not show obvious differences regarding the simulated rainfall. In order to improve the calculation efficiency for further hydrological use (flood warning), we chose to spend 6 hours to spin-up the model.

Givati A, Lynn B, Liu Y, Rimmer A. Using the WRF Model in an Operational Streamflow Forecast System for the Jordan River. Journal of Applied Meteorology and Climatology, 2012, 51(2):285-299.

Hu X M, Nielsengammon J W, Zhang F. Evaluation of Three Planetary Boundary Layer Schemes in the WRF Model. Journal of Applied Meteorology and Climatology, 2010, 49(9):1831-1844.

Wang S, Yu E, Wang H. A Simulation Study of a Heavy Rainfall Process the Yangtze River Valley Using the Two-Way Nesting Approach. Advances in Atmospheric Sciences, 2012, 29(4):731-743.

Point 2: The critical value of Cv is 0.4 and 1.0 for evaluating the rainfall evenness. Please explain how they are derived.

Reply: In order to learn the spatial and temporal evenness of the rainfall in the two study catchment, both spatial and temporal Cv of the storm events from 1985 to 2015 are all calculated. In reality, rainfall in Northern China is much more uneven than the south and it is impossible to find absolute even rainfall in both space and time. So we chose a threshold of 5%, which is also considered in other statistical analyses in the same area, as the critical value to separate even and uneven rainfall events. With the threshold, we found the two critical values of 0.4 for the spatial Cv and 0.6 for the temporal Cv. That is to say, the storm events with the spatial Cv below 0.4 or with the temporal Cv below 1.0 account for 5% of the total storm events from 1985 to 2015 in the study area. Explanations will be added in the manuscript to clarify this issue.

Point 3: At fine spatial simulation resolutions (as pointed out by Referee #1), WRF is effectively running at CPM mode (Convection-permitting model) in which the dynamics of atmospheric convection is treated with sufficient accuracy in order to make it viable to switch off convection parametrization. It would be interesting to run your WRF model again without Cumulus parameterization and compare the results with the Cumulus parameterizations.

Reply: Agreed and we would like to carry out additional runs by masking the cumulus parameterizations. In the case, four members will be added in the physical ensemble in Table 1.

Point 4: It would be helpful to know if any of the six rainfall events have caused any floods in the two study sites. Please explain which WRF simulations are useful to the flood warning purposes, and which are not (ultimately, this is the main goal of WRF applications). Do different parameterizations make any differences for warning purposes?

Reply: Thanks for the referee's suggestion. We would like to provide detailed information of the hydrological responses caused by the six storm events in this study. For example, Event V which occurred on 21 July 2012 has caused the greatest flood during the past 10 year in Jing-Jin-Ji (Beijing-Tianjin-Hebei) area and received widespread attention in China. The 24 h rainfall accumulation was 155.43 mm in Zijingguan catchment and the peak flow reached 2580 m3/s at the catchment outlet. In such cases, accurate rainfall simulations and predictions can do great help to flood warning. However, to analyze the usefulness of the WRF simulations to flood warning, the rainfall-runoff transformation processes should be further considered. This will involve many uncertainties, such as the choice of the rainfall-runoff model, the data used for model calibration, and the involvement of a real-time updating scheme, etc., which also have considerable impact on the accuracy of the flood forecasting results. In this study, the physical ensembles work as a whole to reduce the uncertainty in rainfall simulation caused by a determined set of parameterisations. The exploration of different parameterisations from the flood warning purposes is an important and interesting issue, and worthy to be discussed in our further study.

Point 5: Language issues: The paper has several typos/grammatical errors. Please go through the whole paper carefully to remove them. For example, " the precipitation easily cause flood...(causes)", " which is trick for forecasting accurately (tricky)", " found by a lot of simulation. . .(simulations) ", etc.

Reply: We will go through the whole paper carefully and make efforts to improve the readability of the paper. Grammar and spelling errors will be corrected in the revision.

[Figure]

**Table 1.** The constitution of the WRF physical ensemble

| Member ID | Microphysics | PBL | Cumulus parameterisation |
|---|---|---|---|
| 1 | Lin | YSU | KF |
| 2 | WSM6 | YSU | KF |
| 3 | Lin | MYJ | KF |
| 4 | WSM6 | MYJ | KF |
| 5 | Lin | YSU | GD |
| 6 | WSM6 | YSU | GD |
| 7 | Lin | MYJ | GD |
| 8 | WSM6 | MYJ | GD |
| 9 | Lin | YSU | BMJ |
| 10 | WSM6 | YSU | BMJ |
| 11 | Lin | MYJ | BMJ |
| 12 | WSM6 | MYJ | BMJ |
| *13* | *Lin* | *YSU* | */* |
| *14* | *WSM6* | *YSU* | */* |
| *15* | *Lin* | *MYJ* | */* |
| *16* | *WSM6* | *MYJ* | */* |

**Fig. 1.**

---

## Short Comment (SC2) · 11 Jan 2017

Physical parameterisations have effect on rainfall simulation of the WRF model. This study provides a reference for choosing the appropriate physical parameterisations to simulate different storm types in Northern China. The verification indices are novel and transferable. The results are also analyzed clearly and in detail. Now I have some detailed questions:

1. page 6, line 21: "In order to compare the simulations for different storm events, the final values of the three continuous indices in both two dimensions are represented as percentages of the corresponding average observations". The equation (7)-(9) should be modified as the sentences above, in two different dimensions.

2. page 5, line 16: Mention of the interpolation from observations using the Thiessen polygon method- no reference to this method.

3. Adding some comments in section 5 Discussion: How the rainfall forecasts influence the flood forecasts.

---

## Author Comment (AC5) · 18 Jan 2017

We thanks for the thoughtful considerations from Professor Chu. We agree with the three comments which may help to improve the paper. The explanations to the comments are showed below point-by-point:

Point 1: page 5, line 16: Mention of the interpolation from observations using the Thiessen polygon method- no reference to this method.

Reply: Two references are added in Line 16 page 5 to indicate the application and advantages of the Thiessen polygon method: ". . . which is calculated by the Thiessen polygon method based on the observations of the rain gauges (Sivapalan and Blöschl,

1998; Jarvis et al, 2013)."

Sivapalan M., Blöschl G., 1998. Transformation of point rainfall to areal rainfall: Intensity-duration-frequency curves. J. Hydrol. 204, 150-167.

Jarvis D., Stoeckl N., Chaiechi T., 2013. Applying econometric techniques to hydrological problems in a large basin: Quantifying the rainfall–discharge relationship in the Burdekin, Queensland, Australia. J. Hydrol. 496, 107–121.

Point 2: Adding some comments in section 5 Discussion: How the rainfall forecasts influence the flood forecasts.

Reply: The following sentences will be added in Line 16, page 11: "An efficient flood forecasting system should be able to provide not only accurate forecasts but also long enough lead times for corresponding actions to be taken. Rainfall forecasts can provide rainfall information in the future, which can be used by rainfall-runoff model to forecast the flood and extend flood forecast period. However, the peak flood, flood peak appearance time, flood process are all significantly influenced by the rainfall accumulations and the spatiotemporal distribution of the rainfall (Schellekens et al, 2011; Cane et al, 2013; Fan et al, 2015). In such cases, the accurate rainfall forecast can do great help to flood warning."

Three references are added:

Schellekens J, Weerts A H, Moore R J, Pierce C E, Hildon S. The use of MOGREPS ensemble rainfall forecasts in operational flood forecasting systems across England and Wales. Advances in Geosciences, 2011, 29, 77-84.

Cane D, Ghigo S, Rabuffetti D, Milelli M. Real-time flood forecasting coupling different postprocessing techniques of precipitation forecast ensembles with a distributed hydrological model. The case study of may 2008 flood in western Piemonte, Italy. Nat. Hazard. Earth Sys., 2013, 13(2), 211-220.

Fan F M, Collischonn W, Quiroz K J, Sorribas M V, Buarque D C, Siqueira V A. Flood

forecasting on the Tocantins River using ensemble rainfall forecasts and real-time satellite rainfall estimates. J. Flood Risk Manag., 2015, 9(3), 278-288.

Point 3: page 6, line 21: "In order to compare the simulations for different storm events, the final values of the three continuous indices in both two dimensions are represented as percentages of the corresponding average observations". The equation (7)-(9) should be modified as the sentences above, in two different dimensions.

Reply: The three equations are modified as:
* * *
$$RMSE = \frac{\sqrt{\frac{1}{M}\sum_{j=1}^{M}\left(P_j - Q_j\right)^2}}{\frac{1}{M}\sum_{j=1}^{M}Q_j} \times 100\%$$ (7)

$$MBE = \frac{\frac{1}{M}\sum_{j=1}^{M}\left(P_j - O_j\right)}{\frac{1}{M}\sum_{j=1}^{M}Q_j} \times 100\%$$ (8)

$$SD = \frac{\sqrt{\frac{1}{M-1}\sum_{j=1}^{M}\left(P_j - O_j - MBE\right)^2}}{\frac{1}{M}\sum_{j=1}^{M}Q_j} \times 100\%$$ (9)

**Fig. 1.** Equations

---

## Author Response (AR1)

**Point 1:** Regarding the question raised by Y. Zhang and Referee #2, the following sentences and references are added in the manuscript (see line 26-31, page 3) to address the "the start time of the model and spin-up" issue.

"The spin-up period is necessary for WRF model to develop the smaller scale convective features and the widely used lengths are 6 hours (Givati et al, 2012), 12 hours (Hu et al, 2010) and 24 hours (Wang et al, 2012). Different spin-up lengths were tried for the six storm events in this study, whereas results did not show obvious differences regarding the simulated rainfall. In order to improve the calculation efficiency for further hydrological use (i.e., flood warning), a 6h period is chosen to spin-up the model. That is to say the start of the model integration is 6 h earlier than the storm start time and the end time of the model integration is consistent with the storm end time."

*Givati, A., Lynn, B., Liu, Y., and Rimmer, A.: Using the WRF Model in an Operational Streamflow Forecast System for the Jordan River, J.Appl. Meteorol. Clim., 51, 285-299, doi:10.1175/JAMC-D-11-082.1, 2012.*

*Hu, X. M., Nielsengammon, J. W., and Zhang F.: Evaluation of three planetary boundary layer schemes in the WRF model. J. Appl. Meteorol. Clim., 49, 1831-1844, doi:10.1175/2010JAMC2432.1, 2010.*

*Wang, S., Yu, E., and Wang, H.: A simulation study of a heavy rainfall process the Yangtze River valley using the two-way nesting approach. Adv. Atmos. Sci., 29, 731-743, doi:10.1007/s00376-012-1176-y, 2012.*

**Point 2:** Regarding the question raised by Y. Zhang, the following sentences are added in the manuscript (see line 24-25, page 3) to address the "integration step of WRF model" issue.

"The integration step of WRF follows the '6×dx' rule where dx is the grid spacing, and the integration step is 6s for innermost domain (Skamarock and Klemp, 2008)."

**Point 3:** Regarding the question raised by Y. Zhang, the following sentences are added in the manuscript (see line 12-13, page 6) to address the "insignificant precipitation" issue.

"It should be mentioned that the insignificant precipitation (less than 0.1 mm/h) is regarded as no rain."

**Point 4:** Regarding the question raised by Y. Zhang, two references are added in the

manuscript (see line 22-23, page 3) to support the "40 vertical layers with 1 km horizontal resolution" issue.

[revised manuscript text omitted]

**Point 9:** Regarding the suggestion raised by Referee #1, "ensemble N" should be modified as "the member N of the ensemble".

The nomenclature is revised accordingly in the manuscript.

**Point 10:** Regarding the suggestion raised by Referee #1, "Plots showing the domains and orography" should be added in the manuscript. The following two figures are added in the manuscript.

[Figure]

**Point 11:** Regarding the question raised by Referee #1, the climatic region should be

clarified clearly.

The two catchments in the study are located in semi-humid climatic region, which has been clarified clearly in the manuscript.

**Point 12:** Regarding the question raised by Referee #1, Fig 1 should show the location of the rain gauges.

The location of rain gauges are added in Fig 1.

[Figure]

**Point 13:** Regarding the question raised by Referee #1 and Referee #2, the following sentences are added in the manuscript (see line 15-21) to address the "Cv" issue.

"In order to learn the spatial and temporal evenness of the rainfall in the two catchments, both spatial and temporal Cv of the storm events from 1985 to 2015 are all calculated. In reality, rainfall in Northern China is much more uneven than the south and it is impossible to find absolute even rainfall in both space and time. So we chose a threshold of 5%, which is also considered in other statistical analyses in the same area, as the

critical value to separate even and uneven rainfall events. With the threshold, we found the two critical values of 0.4 for the spatial Cv and 0.6 for the temporal Cv. That is to say, the storm events with the spatial Cv below 0.4 or with the temporal Cv below 1.0 account for 5% of the total storm events from 1985 to 2015 in the study area."